# The Green Business and Sustainable Development School—A Case Study for an Innovative Educational Concept to Prevent Big Ideas from Failure

**Jan Nicolai Hennemann** [1,*] **, Bernd Draser** [2] **and Katarina Repkova Stofkova** [3]

1 Department of Communication, University of Zilina, Sweco GmbH, Graeffstr. 5, 50823 Cologne, Germany
2 Ecosign, Akademie für Gestaltung, Vogelsangerstr. 250, 50825 Cologne, Germany; bernd.draser@ecosign.net
3 Department of Communication, University of Zilina, Univerzitná 1, 01026 Žilina, Slovakia; katarina.stofkova@fpedas.uniza.sk
* Correspondence: nicolai.hennemann@sweco-gmbh.de; Tel.: +49-151-50-70-40-15

**Abstract:** This article addresses the question of why initiatives in the field of green business and sustainable development often fail. Therefore, it dismantles some typical patterns of failure and shows—as a case study—how these patterns can be challenged through an innovative educational concept: the green business and sustainable development school. The applied methodology is a real-life project that is designed through methodological elements stemming from business model canvas, theory U, stakeholder participation, and design thinking. The results of the school initiative are discussed and evaluated by four distinctive stakeholder groups and the school's supporting potential to overcome typical patterns of failure in the green business and sustainable development arena by the younger generation in the future is outlined. This article concludes with ideas to enhance the school concept to reach even more stakeholder-groups and increase its reliability and viability.

**Keywords:** educational concept; green business school; "new green deal"; "European green deal"; interdisciplinary capacity and movement building; green failure; young generation collaboration network; prevent big ideas from failure; theory U; science and action-based research; design thinking

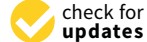


## 1. Introduction

This article presents "The Green Business and Sustainable Development School" (in the following: "the school") as a case study for the implementation of an innovative educational concept to prevent big ideas from failure and to activate the youth through education and training as proclaimed in the European Commission's communication "The European Green Deal" [1] (pp. 18–20). The authors yield more than ten years of experience in designing, executing, and improving school programs on green business and sustainable development with more than 15 European universities and multiple key partners and stakeholders from public and private companies, ministries, universities, research-institutes, think-tanks, NGOs, foundations, associations, municipalities, and finally civil society.

The school initiative aims at three fundamental goals. First, the school seeks to provide students with an (international) networking and learning platform with numerous actors from a wide range of disciplines and levels of experience. Second, professors, lecturers, and researchers will find a broad-based, low-threshold forum to share and further develop new research results as well as innovative teaching concepts with a highly motivated, critical-thinking, and diverse audience. Third, specialists and generalists from business and government can expect a fresh exchange at the highest level. In addition, they can present their companies and organizations and thus draw attention to themselves both in terms of content and as employers. All this happens with a special focus on the topics of green business and sustainable development and enables challenging knowledge and experience curves on all and very different levels.

In order to outline the school structure and operating principles, the paper consists of five sections. The first section describes, as a theoretical background, why ideas and initiatives in the field of green business and sustainable development often fail and what patterns of failure need to be overcome. The second section explores the question of which steps have proven helpful in planning, implementing, and improving schools over the last decade to challenge the patterns of failure. These steps are presented to the reader in a step-by-step do-it-yourself toolkit providing materials and methodology. In a third section, the vision and mission of the school concept is described and the connection to the patterns of failure is revealed. In a fourth section, the authors will disclose structured, multi-layer feedback and learning curves including limitations derived from evaluations and discussions with four distinctive stakeholder groups. The last section of this article investigates the future and presents opportunities for expansion of the green business and sustainable development school. In this section the authors also give hints on quantitative and qualitative research that should be carried out in the future in order to further develop the green business and sustainable development school concept.

## 2. Patterns of Failure—A Theoretical Background

Big ideas and initiatives in the field of green business and sustainable development often fail [2] (pp. 361–370). In order to make the younger generation's initiatives in the field of green business and sustainable development and the school itself more robust and resilient, the authors of this article have started to design and execute green business and sustainable development school programs for multiple institutions and disciplines. All these programs had in common was that they focused on increasing the width and depth of green business and sustainable development related understanding and know-how of all participating individuals and partner organizations.

In a pre-planning phase of the green business and sustainable development school initiative, key reasons were identified as to why the above-mentioned failure is so common. The following five reasons were distilled by literature review and exploratory interviews with experts from industry, associations, universities, ministries, public institutions, and agencies:

1. Green business and sustainable development initiatives are multidimensional and interdisciplinary. Usually they include a technological dimension, an economic dimension, an institutional dimension, and a cultural dimension [1] (p. 3), [3] (pp. 39–40), [4] (pp. 155–183), [5] (pp. 301 et seqq.). In connection with these dimensions many interests and positions of multiple stakeholders need to be taken into consideration.

Not analyzing the underlying systems composed of dimensions, interests, positions, and people can lead to misdiagnosis and misinterpretations that ultimately bear the seeds of failure. On the contrary, if the green business and sustainable development school initiative creates an environment for participants to perceive, experience, and learn to implement collaboration, then everyone involved will quickly realize the added value and potential of such an initiative. The overarching frame is then changed from an "either or" to a "both as well". This should make it possible to understand and implement the multidimensional and interdisciplinary dimensions no longer as a necessity, but as a virtue.

2. Green business and sustainable development initiatives involve inhomogeneous (small) stakeholder groups and individuals [6] (pp. 358–371), [7] (p. 11), [8] (p. 49), [9] (p. 61). Contrary to other industries with a history of several decades to centuries, the field of green business and sustainable development is a comparatively young industry with unsettled structures, finance, and framework conditions.

In this respect global, bird's-eye-view framing, and top-down approaches can lead to gridlock and failure [2] (pp. 362–363). Given that, these approaches often result in abstractions and simplifications. Thus, many local and regional issues and individual ideas are lost or remain unspoken. This leaves the younger generations demotivated and clueless instead of embracing their potential to develop new solutions for existing problems

with fresh eyes and unbiased attitudes. At this point, Margaret Mead's quote holds much wisdom and insight: "Never doubt that a small group of committed people can change the world—in fact, it's the only way the world has ever been changed." [10] (p. 193).

However, top-down approaches should not be questioned per se. Rather, there are challenges such as climate change that can only be effectively addressed on a global scale. The dualism of top-down and bottom-up approaches can be used or even aimed at by school initiatives that cooperate with international partner initiatives in other countries and regions to gradually expand their own outreach and depth. This momentum once set in motion can catalyze the maturation and coalescence of green business and sustainable development initiatives and structures, providing strong growth impetus for the young and emerging industry.

3.    Green business and sustainability (school) initiatives have had a reputation for creating only intangible benefits with little or no visibility. As a result, they are seldom among the top priorities of individuals, institutions, or organizations and must first gain an identity [9] (p. 27), [11] (pp. 206–234), [12] (pp. 1–2, 12–14). This initial situation resonates with the well-known tragedy of the commons as well as with lacking internalization of externalities.

The tragedy of the commons is one crucial factor why humanity makes no serious inroads in averting climate change and investing into green business and sustainable development initiates on much higher scale. [13] (pp. 1–2). This phenomenon plays a major role in environmental sociology and economics and describes the question of whether the behavioral change of one group can have a positive impact at all if all others continue using a common resource as they did before or even reduce their own efforts to reduce their usage as a result. [14] (pp. 224–228).

A second key factor in why humanity is not making serious progress in averting climate change and investing much more heavily in green business and sustainable development initiatives is a reluctance to internalize external costs. The problem of internalization of externalities was most recently exemplified with regards to the German agriculture using nitrate [15] (pp. 1 et seqq.). This phenomenon plays a major role in environmental economics and describes a situation in which the environmental costs resulting from, for example, the production of individual products, are born by the society as a whole. [14] (pp. 198–201). By transferring the costs to the whole society, the originator of the costs is not or hardly incentivized to minimize the costs he causes [14] (pp. 198–201).

Such a situation in the arena in which the school initiative tries to originate and create value can certainly paint a deterrent picture and thus contribute to the failure of initiatives at an early stage.

4.    Green business and sustainable development initiatives experience insufficient funding [16] (p. 9), [17] (p. 1). Even though green products and services have been under development for the last couple of decades, the so-called green sector is still in its nascent stage in terms of commercialization and market acceptance [18] (p. 931).

Hence, mobilization of sustainable investment from all sources is needed [17] (pp. 4 et seqq.). Or, as Otto Scharmer at MIT declares, "We have a system that accumulates oversupply of money in areas that produce high financial and low environmental and social returns, while at the same an undersupply of money in areas that serve important societal investment needs." [19] (p. 13), [10] (pp. 81–83).

5.    Green business and sustainable development initiatives often rely more on intrinsic, self-driven motivation of individuals than on embedded capacities and value creation within organizations and systems [20] (pp. 147–156), [21] (pp. 61–162), [22] (pp. 11–13).

In the longer term, there are hardly any initiatives, let alone entire industries, that can be based and built on intrinsic, nonprofit motivation alone. Sooner or later, the people involved must earn money with their activities and create value in order to be sustainable. Additionally, the initiative itself also consumes monetary values. In fact,

a study conducted by Yale School of Management and the Goldman Sachs Foundation Partnership on Nonprofit Ventures showed that (nonprofit) initiatives pursuing earned-income activities have more employees, believe that they are entrepreneurial, start their business earlier, operate on larger budgets, fund other initiatives and programs, and have a strong desire to see their ventures grow and even replicate [23] (pp. 3 et seqq.), [24] (pp. 225–226). The business model in which initiatives in the field of green business and sustainable development are carried out through volunteerism, donations, and intrinsic motivation should therefore by no means be maintained beyond the initial start-up phase.

In a next step the awareness of these patterns of failure was used to set up a business model canvas (see Figure 1) [25] (pp. 16 et seqq.), [26] (pp. 80–81). With this canvas the school initiators identified and clustered key stakeholders who were eligible and willing to join and contribute to the green business and sustainable development school initiative. Through this process public and private companies, ministries, universities, research-institutes, think-tanks, NGOs, foundations, associations, municipalities, and finally civil society became visible to overcome the identified patterns of failure. They stem from a variety of backgrounds and industries. The next section will present how key partners, who became members and delegates of a core team, commenced to plan, implement, and improve the green business and sustainable development school.

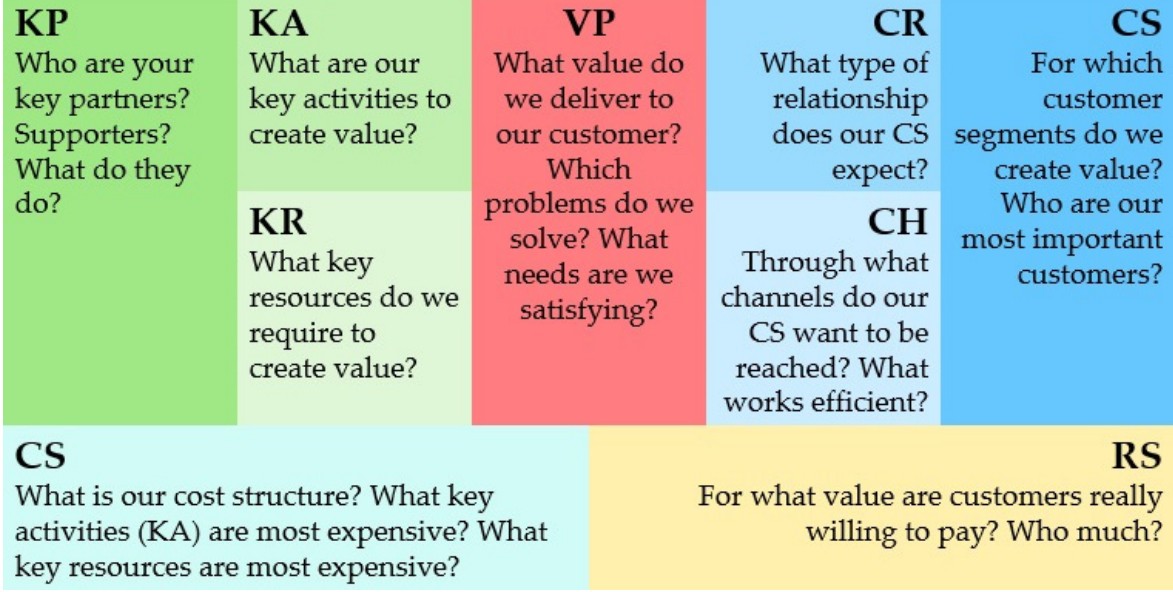

**Figure 1.** Business model canvas to identify key partners (KP) with key activities (KA), value propositions (VP), and relationships (CR) (based on [25]). This work is licensed under the Creative Commons Attribution-Share Alike 3.0 Unported License.

### 3. Materials Preparing the Green Business and Sustainable Development School Through a Step-by-Step Do-It-Yourself Methodology

After a kick-off meeting of the core team that consisted of seven members and delegates from key partners the core team determined concrete steps to plan, implement, and improve a green business and sustainable development school. These steps mature over time and have been structured using the "Theory U" approach depicted below (see below, Figure 2) [10] and [27]. The advantage of this approach is that the "Theory U" process encompassed all phases from idea generation, through planning and implementation, to revision of the school from the outset. Additionally, the underlying toolkit was designed to balance major setbacks (e.g., the unforeseeable departure of one or more key partners, or a COVID-19 pandemic) through preinstalled adaptation loops.

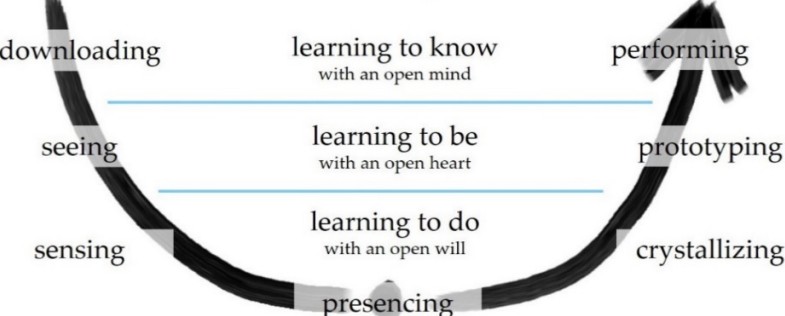

**Figure 2.** The "Theory U" approach to plan and implement the school process (based on [10,28]. This work is licensed under the Creative Commons Attribution-Share Alike 3.0 Unported License).

Downloading: This step helped the core team to overcome the blind spot of an interdisciplinary, fragmented, and yet highly experienced skillset of its members. To prevent team members from not sharing their experiences with a randomly assembled running mates, a focused approach that values the individuality and interdisciplinarity of each core team member was needed. This goal was reached through a journaling practice [10] (pp. 115–124), [29], [30] (pp. 216 et seqq.). A journaling practice sets up a quiet and protected space that allows each journaling participant to enter a process of self-reflection without an obligation to unveil respective answers. Participants were invited to journal their individual thoughts and experiences guided by the following questions.

- Footprints: What have you noticed about yourself that can support a green business and sustainable development school?
- Who have been your mentors and helpers in your personal and professional journey so far?
- Bird's eye: Watch yourself from above. What can you do to make the green business and sustainable development school a success?
- Fast-forward: Imagine you could fast-forward to the very last moments of your professional tenure. Now look back on your professional journey. What would you want to see at that moment? How would you like to design the green business and sustainable development school? What advice do you give your current self that is now about to design the school?
- Crystallization: Now return to the present and crystallize what it is that you want to create: your vision and intention for the next three to five years for the green business and sustainable development school. Describe and draw as concretely as possible the images and elements that occur to you.
- Prototyping: Over next three months. What would your protype of the green business and sustainable development school look like? Who can help you to make your highest aspiration regarding the green business and sustainable development school a reality? What practical first steps would you take over the next three days?

This step welcomes core team members to become aware of their prior experiences and expectations in a safeguarded setting, to articulate them for themselves, and then gradually share their points with the entire team. It laid the foundation for further action, in which the core team was prompted to approach the initiative with fresh eyes.

Seeing: This step is straightforward, although not always easy [10] (pp. 127–138). It is about shifting from the center of an organization or project, leaving its habits and routines to the periphery, the edge of an organization's or project's boundary. Therefore, the following principles were applied:

Members of the green business and sustainable development school core team were asked to question and clarify their individual and organizational intent joining the green business and sustainable development school initiative.

In a second phase team members moved towards the contexts that matter most when conducting the green business and sustainable development school for themselves, for their prospective students, for the industry, and for all other stakeholders they wanted to address and activate.

Last but not least the core team members were asked to suspend their judgement and positions to connect with and aim for the best green business and sustainable development school experience imaginable with value propositions they had already traced during the business model canvas process (see above, Figure 1).

With this step core team members started to open their mind to gain a wider idea of the goals and effects of a successful school. This is an interim goal that can hardly be underestimated in order to successfully drive through the next steps.

Sensing: Members of the core team moved their focus from individual thinking (looking at the green business and sustainable development school from their individual or organizational viewpoint) beyond the organizational and project boundary towards a wider system view [10] (pp. 139–159), [31] (pp. 11–34). The core team members began to realize their relationship towards the system of a green business and sustainable development school and how they collectively could shape it. Four main principles came into play when entering the collective field of sensing [10] (p. 146):

- Charging the container: This tool stems from design thinking processes. A physical box or container is designed by core team members [25] (pp. 541–542), [26] (pp. 104–125), [32] (forthcoming), [33] (pp. 106 et seqq.). Therefore, the container provides a physical object, where members of the core teams craft images of the green business and sustainable development school in delimited spaces. What shall be created? In a time space participants create a timeline in which an agenda and milestones of the process flow into. The relational space holds (personal) relationship with each team member, clears roles, and specifies the infrastructure used in the process.
- Deep diving: This is the gateway towards a success story. Members of the core team become the user of the green business and sustainable development school [10] (pp. 146–148), [34] (pp. 9–12, 19–42). It is about living in the green business and sustainable development school—and collecting (long-time, life-changing) experiences with it. This is much more than reflecting on or creating dialogue with multiple stakeholders.
- Redirecting attention: This step was reached when team members answered that the picture of the whole green business and sustainable development school included themselves when looking at it. Then perception happens from the whole, integrated field (with intention, body and mind), instead within a single and separate actor.
- Open to enact through the heart: As UC Berkeley cognitive psychologist Eleanor Rosch states, this is not a sentimentality or an emotionality but a crucial shift into a deeper field of understanding and enactment for driving success. Almost always, when such a shift happens, there is often a moment of deep silence and/or a question that comes straight from the heart combined with a sigh of relief [10] (p. 148).

With this step core team members became committed and compassionate to the common school initiative, building up trust towards each other. This is the starting point to establish collective will and power of an interdisciplinary, fragmented, and yet highly experienced core team.

Presencing marks the bottom of the "Theory U" process. It is a blend of sensing and presence and marks the crucial "going through the eye of the needle" and "point of no return" moment of a project like the school initiative. To enable core team members to make a meaningful decision in their current situation they need to look at it from the emerging and envisioned future [10] (pp. 161–185). In a nutshell, presencing consists of two shifts: The first shift (place) asks team members to step out of the core team perspective and to look at the school initiative from the outside. The second shift (time) calls the core team members to fast-forward into the future and look back to the initiative describing the perception from there. Three principles accompany these shifts:

- Letting go and surrender—this means that team members need to drop everything from their schedule that is not essential to the school initiative. Questions like: "Does the school allocate credits (ECTS) to the students? Will school partners create journal publications about the school initiative? Will lecturers reduce their teaching load through their school commitment?" were suspended from the agenda as the core team members classified these questions as not essential for the school process. In design thinking this is also known as crafting the minimum viable product (MVP)—something to test fundamental real-world or market hypotheses [35] (p. 76). A school that offers enough features will be useful for attracted students who can then provide feedback for the future [36] (pp. 92 et seqq.), [37] (p. 176–177).

- Inversion describes a team turning the inside out and the outside in. In practice the core team members stopped asking the question of what the school costs are (in terms of individual effort and institutional money). Rather they focused on why and how the school could ease academic processes and will earn money in the future (e.g., through corporate and third-party funding). Another inversion was: "The school hinders us from taking on other projects" to "The school enables us to see and take on other projects".

- Another principle focuses on the power of place (also known as "war room") [10] (p. 182). This should not be underestimated. It is about leveraging the presence and power of certain places. For the core team members this place was a meeting room in the ministry for the environment—as a neutral, safe ground. Every six weeks the core team met there for three hours to proceed the school initiative. Sometimes—due to the COVID-19 pandemic—these meetings took place on a virtual platform, sometimes in a hybrid format with some team members meeting in the ministry whereas others were hooked up via teleconference-tools.

With this setup the core team members were eligible to work themselves through the eye-of-the needle and the task of presencing.

The next step deals with moving up the slope and is called crystallizing. Crystallizing means clarifying vision and intention [10] (pp. 188–193). To enter this step, it is decisive for the members of the core team to have gone through the presencing, eye-of-the needle process. Up to this point they have only felt the possibility of the future with a school. After a presencing moment team members are now poised to bring the conceived individual and collective potential into reality. From now on the notion shifts from "We can't do it" to "We can't not do it". The first step in this upward movement is to put the vision and intention into more clear and specific language. The following two principles come into play when stepping into the crystallizing process:

- Embracing the power of intention of the core team is crucial for the crystallizing process. Keep in mind Margaret Mead's quote, "Never doubt that a small group of thoughtful, committed participants can change the world. Indeed, it is thing that ever has." [10] (p. 193).

- Letting come is the other side of the power of intention. In other words, it means to turn to something new requires the core team to let go something old, for example, an incumbent project. That is a reality. Most members of the core team had to quit other "stuff" to have time and resources for the new school initiative. Since the "U" process had started with observations, team members now had the certainty of being able to make the right decisions as part of their reflection.

The crystallizing process paved the way towards crafting a prototype of the school as the vision and intention of the future keeps evolving.

Prototyping: Having gone through the eye-of-the-needle and having clarified a sense of the future of the school initiative led the core team members into doing, which is also known as prototyping. Prototyping describes the first step into the real world through experimentation [10] (p. 201). The design industry incorporates a long history of prototyping as a necessity to endure relentless competition for ideas. Often this is summarized as "Fail

often, fail early to succeed sooner" [38] (p. 232). It means to present a concept before it is perfected to allow fast-cycle feedback learning and adaptation.

Fail early and learn quickly: The core team members aimed to develop a summer school that would provide in a first step 80 students with an interdisciplinary and international background a school platform over a two-week period. For this purpose, a prototype in form of an MVP was created within three months, which already contained the essential cornerstones of a school, but offered a program over a significantly shorter period (three days). The goal behind this was to realize initial experiences and learning curves, which would then be considered in a larger, full-scale school run-up.

As an MVP prototype of the school, the "Winter School 2019 on Green Business and Sustainability" dedicated three consecutive days to the interfaces between ecology and economy, opened transdisciplinary perspectives to participants, presented current developments in science, design and business, and encouraged interdisciplinary discourse. The School was held at the German Aerospace Center (DLR) in Cologne, the resources competence center: metabolon in Lindlar, the Zeche Zollverein in Essen and the Green Business Summit at Messe Essen trade fair.

Performing: After the successful prototype in 2019, the experience and learning curves were integrated into the steps leading to the school of 2020. Figure 3 shows the participants' field of study during the in-person winter school 2019 and virtual summer school 2020. In total within 12 months more than 160 participants joined the school.

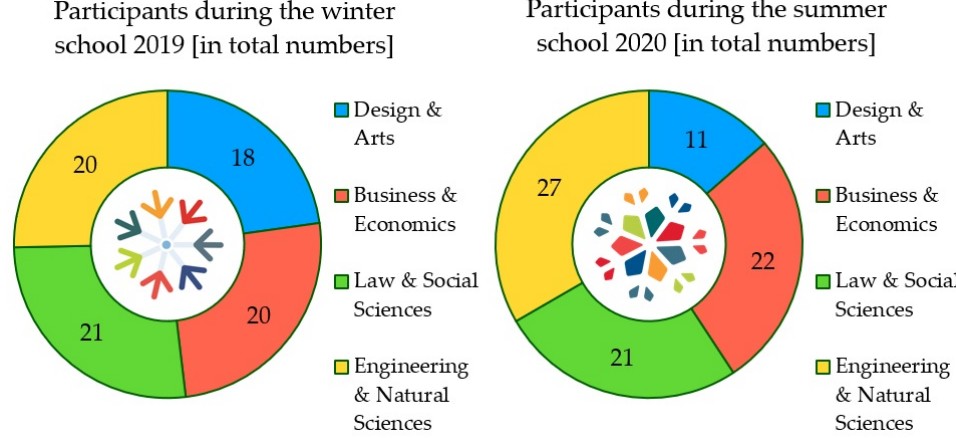

**Figure 3.** Participants' field of study during the in-person winter school 2019 and virtual summer school 2020.

What started in 2019 with a three-day, single-aisle program was developed in 2020 into a two-week program shown in Figure 4. On the left side of each page of the program participants found chronological lectures to multiple topics related to green business and sustainable development. On the right side of each page of the program four parallel workshop-streams on water, urbanity, light, and energy were established. Participants were eligible to choose their favorite workshop-stream during the school and work together with their school mentors and industry experts to solve real-world challenges.

It should be noted that due to the COVID-19 pandemic, the core team had to rewrite the school program into a virtual, two-week format under intense time pressure. The importance of operating the school initiative from the larger eco-system as opposed to operating from each institution's ego-system [10] (pp. 218–2020) became obvious at this point. In order to live and breathe, the initiative emerged into a web of relationships connecting public and private companies, ministries, universities, research-institutes, think-tanks, NGOs, foundations, associations, municipalities, and finally civil society. Remembering that exactly this was the starting point where the school initiative began (see above Section 1. Introduction) marked the starting point of a new and revised "Theory U" process to shape the school of 2021.

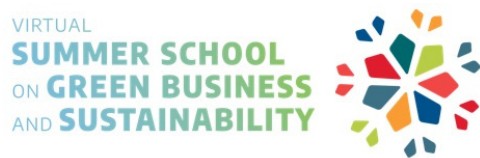

**TIMETABLE:**
**24.8.2020 - 4.9.2020**

| Weekday | Week 1 | | Week 2 | |
|---|---|---|---|---|
| | Morning | Afternoon | Morning | Afternoon |
| MON | Welcome | Freetime | Water Keynotes | Workshops |
| TUE | Energy Keynotes | Kick-Off Workshops | Open Forum | Workshops |
| WED | Circularity Keynotes | Workshops | Social Keynotes | Creative-Time |
| THU | Urbanity Keynotes | Workshops | Student Workshop Presentations | Student Workshop Presentations |
| FRI | Pulse-Check | Half-time Feedback | Student Workshop Presentations | Farewell, Final Feedback |

**Figure 4.** The school's timetable (schematic, German, only) [39,40].

### 4. Connecting the Toolkit to the Patterns of Failure

The previous section of this paper revealed the school's concept and the toolkit which was used to bring the school to life. In this third section the authors will now connect the "tools" addressed in Section 2 with the "patterns of failure" described in Section 1 (see below Table 1), to deepen the understanding about the underlying mechanisms operating "behind the scenes" of the school initiative.

**Table 1.** Relationships between "tools", "patterns of failure", and "mechanisms behind the scenes".

| "Tools" Used to . . . | . . . Challenge and Change These "Patterns of Failure" by . . . | . . . Applying These Mechanisms and Operations "Behind the Scenes" |
|---|---|---|
| Downloading | → multidimensional, complex and interdisciplinary patterns <br> → self-driven motivation patterns | The downloading process opens a safe space for core team members right from the start. This allows them to work on common profiles and blind spots and thus explore their interdisciplinarity in the best possible way to make collaboration a motivating factor for the school initiative. |
| Seeing | → inhomogeneous, individual stakeholder patterns <br> → intangible benefits patterns | During this step core team members question and clarify their individual and organizational intent. They receive a notion of what matters most for them and for the others. This helps to suspend judgement and positions in order to open one minds for the school initiative and gain a wider idea of what is possible. |
| Sensing | → inhomogeneous, individual stakeholder patterns <br> → self-driven, intrinsic motivation patterns | The sensing phase intensifies the core team members' relationship to the entire system in which the school initiative is embedded. It builds up trust and marks the starting point of a collective will and power to act with commitment and compassion. |
| Presencing | → intangible benefits patterns <br> → insufficient, dislocated funding patterns <br> → self-driven, intrinsic motivation patterns | This step can also be described as going through the eye of the needle—a point of no return. Core team members focus on the minimum viable product to focus themselves to what is necessary throughout the next stages. |
| Crystallizing | → all patterns | With this step core team members step into a process of formulating and carrying out what has been conceived and developed during the before-mentioned phases. The attitude shifts from "we can't do it" to "we can't not do it". |
| Prototyping | → all patterns | This phase leads core team members into doing. A culture of "fail often and fail early" can emerge. |
| Performing | → all patterns | During the performing process a viable prototype is taken to the next level. |

For future initiatives which intend to use this case study and toolkit it should be noted that the above mentioned "tools", "patterns of failure", and "mechanisms behind the scenes" do neither necessarily occur in every initiative nor can be solved in the same way that is depicted in this article. However, the authors, who hold an extensive and long experience with such initiatives, confess that this article reveals several patterns and repetitions they experienced so far.

## 5. Discussion and Evaluation

In this section, the authors will disclose structured, multi-layer feedback [41] (pp. 56 et seqq.) and learning curves from evaluations and discussions with four distinctive stakeholder groups to detect limitations of the green business and sustainable development school initiative:

a.　participating students and researchers
b.　members of the core team
c.　key partners and key stakeholders (ministries, companies, universities, and institutes)
d.　a broader expert audience (e.g., participants of industry workshops and conferences).

Participating students and researchers were asked three times for their feedback.

The first feedback took place as an open, oral discussion after the first week of the school (half-time feedback in a so-called "room for wishes"). In this "room for wishes" the core team asked participants for spontaneous ideas for program changes and additions for the second week. Participating students actively accepted this offer. Thus, an open exchange forum for students was established (a room without the presence and supervision of core team members), a book presentation by a participant who had published her first book shortly before the school took place, and thirdly an exercise in mindfulness was held by an experienced yoga master in the second week of the school.

A second oral feedback opportunity was set up at the end of the school program ("farewell feedback"). In a dialogue format between core team members and school participants and researchers the following five points shall be highlighted here: First, the respondents stated that the inclusive and collaborative school approach sparked high potential for individual and joint progress in the future. Secondly, as the School offered a rather long and intensive virtual program, which was excellently visualized in the timetable shown in Figure 4, it was helpful or necessary that the core team repeatedly addressed the different agenda items in its communication towards all participants. Thirdly, the mixture of knowledge transfer in a lecture format in the morning and idea creation in different workshop formats in the afternoon, which was constantly stretched over two weeks, was highly praised by the participants. The fourth feedback item expressed the students' wish for a follow-up to put the results of the workshop streams into practice, preferably directly linked to a real financing opportunity, for example, granted by a state finance agency such as the NRW Bank. Finally, the students noted in their "farewell feedback" that a two-week program, comprising a workload of more than 60 h, should become eligible to credits in terms of ECTS. The school's certificate of attendance is a good first step. However, the students' effort and commitment should be made more rewarding in real ECTS impacting their study progress.

In addition to the two oral feedback opportunities, a feedback questionnaire was sent out after the school to all participants asking these questions: To which workshops did you enroll? What did you like about your workshops and what suggestions for improvement do you have? Do you have any other comments or feedback about the entire program? How did you like the school in general (e.g., regarding organization, moderation, lectures, structure, workload)? What can we do better next time? On a scale from 1 to 10 (1 is not likely, 10 is most likely): Would you recommend the school to others? On average the school received a nine out of ten regarding the latter question.

However, the overall tone of the participants' feedback was that the most important take-away from the school was the interdisciplinary experience students and researchers had during the program. Most participants noticed that implementing their ideas takes more than their own expertise and excellence in their respective field of study. Rather, they

gained the confidence to use the school's network to implement their projects and start-up ideas more efficiently, for example in the field of natural sciences, through good communication design [42] (pp. 259–266). With that the participant-related fundamental goal of the school was met: the school demonstrably provided students with an (international) networking and learning platform with numerous actors from a wide range of disciplines and experiences.

Unlike the participants, the members of the core team met for a two-hour review workshop to generate their feedback regarding the school initiative. In this workshop three straightforward questions were asked: What worked well? What didn't work so well? What does the next school need to do to increase its value proposition? The following three points seemed most significant to the core team members: Firstly, most core team members admitted that they underestimated two things. On the one hand, the commitment and excellence of the participating students (most of them spent extra hours, exceeding the planned workshop hours). On the other hand, they underestimated the re-planning effort from transcribing the school from an in-person format into a pandemic-safe virtual format. Secondly, most core team members reflected on the students' desire for ECTS eligibility and committed themselves to examine with their respective university administration under which circumstances ECTS can be granted to students. From their eyes, the students' desire can be fully understood. Third, as successful school initiatives have been planned and implemented repeatedly, the core team members and adjacent partners feel equipped to approach and involve additional supporters in the run-up to the next school to strategize lectures and workshop formats in an increasingly integrative way. Here, core team members, stemming from academic institutions, see their greatest personal and professional benefit. If they succeed in placing their own scientific and research activities in the field of green business and sustainable development with corporate partners and students on a broader, more effective basis right from the start, their results may become more resilient and effective. With that the professor- and lecturer-related fundamental goal of the school was met: the school demonstrably provided professors, lecturers and researchers with a broad-based, low-threshold forum to share and further develop new research results as well as innovative teaching concepts with a highly motivated, critical-thinking, and diverse audience.

Key partners and key stakeholders received feedback calls to reflect on their individual impressions. The following questions were asked: How did you feel about the School? What worked well from your point of view or from the point of view of those you talked to about the school? What did not work so well from your point of view or from the point of view of those you talked to about the school? What wishes do you, or third parties with whom you have spoken about the school, articulate for the next school? On a scale from 1 to 10 (1 is not likely, 10 is most likely): Would you recommend the school to others? On average the school received a nine out of ten regarding the latter question. Additionally, the following three recommendations seemed most significant to key partners and key stakeholders: First, key partners and key stakeholders pursue multi-layered goals by participating in the school initiative. These goals range from public relations and employee recruitment to testing and developing product ideas and solutions. Secondly, in order to achieve these goals in the best possible way, key partners and key stakeholders need to be involved in the planning of the school at an early stage and have enough time to approach their internal processes professionally. Thirdly, key partners and key stakeholders are aware that early involvement and coordination in the planning process of the school generates additional effort. They are willing to compensate for this additional effort. To this end, there should be a coordination meeting in advance of further school initiatives in order to achieve win-win solutions for all parties involved. With that the partner and key stakeholder-related fundamental goal of the school was met: the school demonstrably provided partners and key stakeholders a fresh exchange at the highest level. They can present their companies and organizations to high potentials and a wider (expert) public and thus draw attention to themselves both in terms of content and as employers. All this

happens with a special focus on the topics of green business and sustainable development and enables challenging knowledge and experience curves on all and very different levels.

Finally, the school initiative was presented and discussed at conferences and workshops with a wider group of experts. They provided recommendations from non-stakeholders as an external perspective. One prominent workshop with a broader and distant group of experts took place during the Bundesverband Mediation Kongress 2020 [43]. There, the following strengths and weaknesses regarding the school initiative were reflected: Green business and sustainable development are on the agenda of most companies and institutions and for most of them the topics are relatively new. A school like the one described can initiate a valuable leverage to the companies' and institutions' efforts towards green business and sustainable development. Summer and winter schools are well established in the international context, especially in the Anglo-Saxon culture. This may be a disadvantage if international students are also to be addressed by this new school initiative.

Finally, it should be noted that the issue needs to be seen in the context of the present, when markets are characterized by strong competition and innovation is an important tool enabling companies to gain a competitive advantage. On the other hand, innovation alone is not enough, but more importance is placed on creating innovations implemented on the principle of sustainable development. Eco-innovation can be understood as the selection of suitable materials, processes, and distribution methods that are used with lower energy and natural resource consumption and, overall, with less impact on the environment [44] (p. 8). From this point of view, the above-mentioned initiatives in the field of green business and sustainable development are very important and require a great deal of attention.

Therefore, if the school's focus can be established and its profile gets sharpened, this initiative can become a flagship in the area of innovation labs, capacity building, and action research. Therefore, the partners need to work on their interests, interdisciplinarity and complementarity. Only if the roles and tasks are clearly identified and divided on a permanent basis can such a joint project flourish for a long period of time.

## 6. Conclusions and Outlook

The initial goal of the school was to create an innovative educational concept that prevented big ideas in the green business and sustainable development arena from failing. This goal was twofold. On the one hand, the school itself was to become an initiative that would not fail. On the other hand, the students and participants of the school should be supported in their initiatives in order to prevent them from failing. Consequently, a methodological concept was used, further developed during the creation of the school, which had a good probability of making the school a success story. This concept was then also made tangible, transparent, and learnable for all interested students, partners, and stakeholders within the school. In other words, participation in the school created initial personal references, and thus ensured real-time access to the concept.

As shown in the previous sections, the underlying concept has been successful. This is not surprising, as the concept itself provides a high degree of adaptability and even major setbacks (e.g., the unforeseeable departure of one or more key partners, or a COVID-19 pandemic) can be balanced through preinstalled adaptation loops. Based on the core team's experience and feedback with four stakeholder-groups outlined above, the following opportunities for expansion and improvement shall be mentioned in this outlook for the school:

- Intensify the virtual and hybrid platform experience to spend time with fellows as if in real life. Such a platform will foster global exchange and outreach through the school initiative minimizing financial and emission implications.
- Establish and maintain a vivid alumni network on social platforms and/or a native school website. The alumni network will stabilize the school initiative and provide valuable supporters and impulse generators from its own ranks for the future and upcoming schools.

- Expanding the feedback framework with quantifiable surveys. So far, the focus has been on qualitative feedback. The larger and thus more representative the number of participants and the number of key partners, the more it makes sense to add quantifiable metrics to the feedback structure in order to advance the value-oriented development of the program for all.
- Initiate a professional recruiting platform to increase both attractiveness for companies, supporters, and sponsors as well as attractiveness for students who seek internships and jobs.
- Create a shuttle (double) loop between, for example, a student summer school (creating ideas and solutions) and a corporate winter school (reflecting ideas and solutions derived from previous summer schools and drafting assignments for upcoming summer schools). Through this added value for all stakeholders, financial independence, and consolidation can be reached through sponsorships and service charges in a next step. In return a real-world knowledge sharing innovation network can evolve [45] (pp. 19–20).

With that the authors express their deep wish that multiple green business and sustainable development school initiatives will sparkle around the world to accompany generations that are needed to solve the most pressing challenges of our time.

**Author Contributions:** Conceptualization, J.N.H., B.D., and K.R.S.; methodology, J.N.H., B.D., and K.R.S.; validation, J.N.H., B.D., and K.R.S.; formal analysis, J.N.H., B.D., and K.R.S.; investigation, J.N.H., B.D., and K.R.S.; resources, J.N.H., B.D., and K.R.S.; writing—original draft, J.N.H., B.D., and K.R.S.; writing—review and editing, J.N.H., B.D., and K.R.S.; visualization, J.N.H., B.D., and K.R.S.; supervision, J.N.H.; project administration, J.N.H., B.D., and K.R.S. All authors have read and agreed to the published version of the manuscript.

**Funding:** This research received no external funding.

**Institutional Review Board Statement:** Not applicable.

**Informed Consent Statement:** Not applicable.

**Data Availability Statement:** Not applicable, see references.

**Acknowledgments:** The green business and sustainable development school initiative is supported by the Ministry for Environment, Agriculture, Conservation and Consumer Protection of the State of North Rhine-Westphalia (MULNV) which is part of the government of the German state of North Rhine-Westphalia. Further acknowledgements go to the partner universities and institutes of the school initiative: the Wuppertal Institute, the ecosign/Akademie fuer Gestalltung, the Joint Centre Urban Systems (JUS) at the University of Duisburg-Essen, the University of Applied Sciences Bochum, the Institute of Energy Economics at the University of Cologne (EWI), the University of Cologne and the Folkwang University of the Arts. This paper was supported by project VEGA 1/0518/19 and KEGA 043ZU-4/2019.

**Conflicts of Interest:** The authors declare no conflict of interest.

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
