# Peer review of "The Green Business and Sustainable Development School—A Case Study for an Innovative Educational Concept to Prevent Big Ideas from Failure"

_sustainability, doi:10.3390/su13041943_

Round 1

Reviewer 1 Report

The aim of the paper is to address the question of why initiatives in the field of green business  and sustainable development often fail. The study of the causes of failure in green business and sustainable development initiatives is an interesting subject nowadays from a scientific and industrial point of view, however; unfortunately, the research developed by the authors presents certain limitations.

1.- Authors are recommended to adapt the structure of the paper to a scientific article structure, including introduction, background, materials, research methods used, results and conclusions. Likewise, the authors are recommended to clearly detail the results of their research in accordance with the objective of obtaining the causes of failure in initiatives of green business and sustainable development.

2.- The authors present the results of their research in a qualitative and descriptive way. Authors are recommended to include quantitative data that structures their research in hypotheses and scientific validation results, in this way it would be possible to rigorously evaluate the impact of their research compared to the research carried out by other authors.

3.- The authors include a study case and a description of the activities carried out. The authors are recommended to include tables with quantitative data regarding the number of participants, institutions of origin, position or function of the participants, time of participation in the sessions, number of sessions per participant, etc.

4.- The information in Figures 1,3, 4 and 5 is not legible. Authors are recommended to include this information in an expanded way in the additional information of the paper or in an annex.

5.- It is recommended to include Figure 2 with a white background.

Author Response

Dear Reviewer 1, first of all, we would like to thank you for your valuable comments and suggestions, which really motivated us to improve the quality of the paper. We also thank you for your assessment that the study is an interesting subject from a scientific and industrial point of view. We also believe that decent work has been done and that this topic is important to share with the readers of the journal. Now let's come step by step to your suggestions for improvement:

  • We adapted the structure of the paper including sections 1. “Introduction”, 2. “Patterns of failure – a theoretical background”, 3. “Materials preparing the green business and sustainable development school through a step-by-step do-it-yourself methodology”, 4. “Connecting the toolkit to the patterns of failure”, 5. ”Discussion and evaluation” and 6. “Conclusions and Outlook”. We added in almost all sections (especially in sections 1, 2, 5 and 6) more details for clarification.
  • So far, the experience and lessons learned from this novel School initiative have been qualitative rather than quantitative. Nevertheless, we have highlighted the quantitative composition of participants of the School in Figure 3. Also the 8 involved institutions Ministry for Environment, Agriculture, Conservation and Consumer Protection of the State of North Rhine-Westphalia (MULNV) which is part of the government of the German state of North Rhine-Westphalia, the Wuppertal Institute, the ecosign/Akademie fuer Gestalltung, the Joint Centre Urban Systems (JUS) at the University of Duisburg-Essen, the University of Applied Sciences Bochum, the Institute of Energy Economics at the University of Cologne (EWI), the University of Cologne and the Folkwang University of the Arts are now listed in the acknowledgement. As the school intends to remain an open, “breathing” and growing initiative the institutions should not be highlighted in the text main body as other institutions who read the paper may get the impression, that this initiative is a closed-shop project. We hope here for your kind understanding.
  • The operative details of the school (duration of sessions, total number of hours spent in the school etc.) are implicitly woven into section 3 (lines 330 et seqq “a two-weeks program shown in Figure 4. On the left side of each page of the program participants found chronological lectures to multiple topic related to green business and sustainable development. On the right side of each page of the program four parallel workshop-streams on water, urbanity, light and energy were established. Participants were eligible to choose their favorite workshop-stream during the school and work together with their school mentors and industry experts to solve real-world challenges.” and section 5 (e.g. lines 376 et seqq.) “the mixture of knowledge transfer in a lecture format in the morning and idea creation in different workshop formats in the afternoon, which was constantly stretched over two weeks, was highly praised by the participants.” […] Finally, the students noted in their “farewell feedback” that a two-week program, comprising a work-load of more than 60 hours, should become eligible to credits in terms of ECTS.”
  • Additionally, operative details can be taken from figure 4 as a rough, visual idea. If you wish more details, we can add the timetable in table 4 as an annex. Therefore, we have to “clean” the timetable you see in figure 4 as it contains personal cell phone numbers of professors and admin staff etc. Also, the keys for still existing Zoom virtual rooms are included. In this respect I hope for your understanding, but I guess the readership gets the idea, that the school took place for two weeks in a row…
  • The quality of the figures was improved, they should now be legible. In Figure 4, however, we do not want to make the readership to read the details. We just want to refer to the visual impression in the text main body. But of course, we can also build a fictive and simplified Figure. However, the majority of the consulted people told us to keep the original impression to give the readership at that point a glimpse into the actual school. But as I said we can also change that Figure.
  • We accepted your challenge to develop the paper forward and hope that we can meet the readerships and your expectations on this case study.

Dear Reviewer 2, first of all, we would like to thank you for your valuable comments and suggestions, which really motivated us to improve the quality of the paper. We also thank you for your assessment that an interesting research work has been done and for your judgment that this topic is important for the readers of the journal. We share your assessment. Now let's come step by step to your suggestions for improvement:

  • In section 1 “Introduction” we included more clearly the aim of the paper as well as the aim of the school.
  • The linkage to “sustainability” is now intensified and woven especially in sections 1 “Introduction”, 2 ”Patterns of failure – a theoretical background”, 5 “Discussion and evaluation” and 6 “Conclusion and outlook”. Additionally, we think, that the linkage is immanent in the topic.
  • The theoretical background now provides an in-depth review. We discussed that point also with all six participating universities and institutions (they are now also listed in the acknowledgement” and they believe, that section 2 ”Patterns of failure – a theoretical background” now carry substantial analysis and insight. So your point, that the original version remained at a modest level, was really fair and helpful.
  • The discussion and implications were also extended in sections 5 “Discussion and evaluation” and 6 “Conclusion and outlook” and we also worked out some more practical and academic implications (especially in section 5 “Discussion and evaluation” where we talk about drivers for (a) (academic) core team members and (b) key (business) partners and key (public) stakeholders.
  • The quality of the figures was improved, the should now be legible. In Figure 4, however, we do not want to make the readership to read the details. We just want to refer to the visual impression in the text main body. But of course we can also build a fictive and simplified Figure. However, the majority of the consulted people told us to keep the original impression to give the readership at that point a glimpse into the actual school. But as I said we can also change that Figure.
  • We also take into account the format and references. Add 2 more references and add eight institutions (ministry, universities and institutions) to the acknowledgement section.
  • We accepted your challenge to develop the paper forward and hope that we can meet the readerships and your expectations on this case study.

Thank you very much.

Reviewer 2 Report

The proposed article studies the important question -why initiatives in the field of green business and sustainable development often fail. From the overall presentation I would say that an interesting research work has been done. The topic is also important for the readers of the journal. However, I have a few more significant challenges with the paper. 

From my point of view, there are some revisions the authors should consider to improve the paper. 

  • The aim of the paper should be included more clearly in the introduction section. 
  • To be consistent with the aims of the Journal, the theme of sustainability must be central or in any case strongly interdependent with the object of the research. Therefore, the authors should include more arguments in order to show the link between the green business and sustainable development school, “innovative educational concept to prevent big ideas from failure” and sustainability.  
  • The theoretical part remains at a modest level. At this stage, it does not yet provide an in-depth review of the previous literature. It is more a description than analysis. Thereforea more detailed explanation of theoretical background and research design needs to be supplemented for this paper to be published. 
  • The original contribution of the research has to be presented by focusing on the research results based on the research questions. 
  • The discussion and implications are rather short and they should be extended. 
  • You need to improve the practical and academic implications. 
  • However, the paper has to underline the limits of the research. 
  • The quality of the figures is not sufficient.  
  • The authors have to take into account the Paper format.  
  • English language and style are fine/minor spell check required [For example, “field of g green business” (page 2)] 
  • The authors have to pay attention to references inside the paper as well as the reference list.  

Please see Instructions for Authors-  “In the text, reference numbers should be placed in square brackets ], and placed before the punctuation; for example [1], [1–3] or [1,3]. For embedded citations in the text with pagination, use both parentheses and brackets to indicate the reference number and page numbers; for example [5] (p. 10). or [6] (pp. 101–105). Journal Articles: 1. Author 1, A.B.; Author 2, C.D. Title of the article.Abbreviated Journal NameYear,Volume, page range.”). 

Revising is always a challenging job, but I think you can develop the paper forward. 

Author Response

Dear Reviewer 1, first of all, we would like to thank you for your valuable comments and suggestions, which really motivated us to improve the quality of the paper. We also thank you for your assessment that an interesting research work has been done and for your judgment that this topic is important for the readers of the journal. We share your assessment. Now let's come step by step to your suggestions for improvement:

  • In section 1 “Introduction” we included more clearly the aim of the paper as well as the aim of the school.
  • The linkage to “sustainability” is now intensified and woven especially in sections 1 “Introduction”, 2 ”Patterns of failure – a theoretical background”, 5 “Discussion and evaluation” and 6 “Conclusion and outlook”. Additionally, we think, that the linkage is immanent in the topic.
  • The theoretical background now provides an in-depth review. We discussed that point also with all six participating universities and institutions (they are now also listed in the acknowledgement” and they believe, that section 2 ”Patterns of failure – a theoretical background” now carry substantial analysis and insight. So your point, that the original version remained at a modest level, was really fair and helpful.
  • The discussion and implications were also extended in sections 5 “Discussion and evaluation” and 6 “Conclusion and outlook” and we also worked out some more practical and academic implications (especially in section 5 “Discussion and evaluation” where we talk about drivers for (a) (academic) core team members and (b) key (business) partners and key (public) stakeholders.
  • The quality of the figures was improved, the should now be legible. In Figure 4, however, we do not want to make the readership to read the details. We just want to refer to the visual impression in the text main body. But of course we can also build a fictive and simplified Figure. However, the majority of the consulted people told us to keep the original impression to give the readership at that point a glimpse into the actual school. But as I said we can also change that Figure.
  • We also take into account the format and references. Add 2 more references and add eight institutions (ministry, universities and institutions) to the acknowledgement section.
  • We accepted your challenge to develop the paper forward and hope that we can meet the readerships and your expectations on this case study.

Round 2

Reviewer 1 Report

The authors have addressed satisfactorily the points raised during the review.Therefore, I recommend the publication of this article.

Author Response

Dear Reviewer 1,

thank you very much for your positive reply. Again we extended and improved the manuscript. Also we improved the quality of figure 4 – in our view all figures are readable now. We also paid attention to the reference list and included some more sources/references and gave several more references in the text main body.

Compared to the initial version the review process really improved the quality of the paper.

Thank you very much for your professional suggestions and comments. We now look forward to the publication of the paper.

Best wishes  

Reviewer 2 Report

Dear Authors, 

In the revised version, the manuscript has been extended and improved and my comments have been covered.  However, the paper has to underline the limits of the research. The quality of the figures is not sufficient. The authors have to pay attention to reference list. 

 Best regards 

Author Response

Dear Reviewer 2,

thank you very much for your positive reply. Again we extended and improved the manuscript. In our view the paper now underline the limits of the research. Also we improved the quality of figure 4 – in our view all figures are readable now. We also paid attention to the reference list and included some more sources/references and gave several more references in the text main body.

We hope our review can satisfy your suggestions. Compared to the initial version the review process really improved the quality of the paper.

Thank you very much for your professional suggestions and comments. We now look forward to the publication of the paper.

Best wishes